# Differential Confocal Optical Probes with Optimized Detection Efficiency and Pearson Correlation Coefficient Strategy Based on the Peak-Clustering Algorithm

**DOI:** 10.3390/mi14061163

**Published:** 2023-05-31

**Authors:** Zhiyi Wang, Tingyu Wang, Yongqiang Yang, Xiaotao Mi, Jianli Wang

**Affiliations:** 1Changchun Institute of Optics, Fine Mechanics and Physics, Chinese Academy of Sciences, Changchun 130033, China; wangzhiyi18@mails.ucas.edu.cn (Z.W.); wangtingyu21@mails.ucas.ac.cn (T.W.); 2College of Materials Science and Opto-Electronic Technology, University of Chinese Academy of Sciences, Beijing 100049, China; mixiaotao@ciomp.ac.cn; 3Jilin Provincial Key Laboratory of Intelligent Wavefront Sensing and Control, Changchun 130033, China

**Keywords:** compensation strategy, differential confocal microscopy, stride length, peak clustering, Pearson correlation coefficient

## Abstract

Quantifying free-form surfaces using differential confocal microscopy can be challenging, as it requires balancing accuracy and efficiency. When the axial scanning mechanism involves sloshing and the measured surface has a finite slope, traditional linear fitting can introduce significant errors. This study introduces a compensation strategy based on Pearson’s correlation coefficient to effectively reduce measurement errors. Additionally, a fast-matching algorithm based on peak clustering was proposed to meet real-time requirements for non-contact probes. To validate the effectiveness of the compensation strategy and matching algorithm, detailed simulations and physical experiments were conducted. The results showed that for a numerical aperture of 0.4 and a depth of slope < 12°, the measurement error was <10 nm, improving the speed of the traditional algorithm system by 83.37%. Furthermore, repeatability and anti-disturbance experiments demonstrated that the proposed compensation strategy is simple, efficient, and robust. Overall, the proposed method has significant potential for application in the realization of high-speed measurements of free-form surfaces.

## 1. Introduction

Free-form surface elements have become increasingly prevalent in modern optical research, industry, and commercial fields due to their high degrees of freedom that can correct different aberrations, satisfy high performance requirements [1], and meet the demands for lightweight and miniaturization in modern optical systems. Since there is no contact between the optical probe and the measured part, there is no stress effect, and the optical probe profiler can achieve higher measurement accuracy without scratching the measured surface. The profilometer utilizes a probe-scanning method to directly test the profile of the measured surface and obtain three-dimensional (3D) profile information for each sampling point while estimating the profile error based on analysis, fitting, and reconstruction. Reflection confocal microscopy is regarded as a mature optical probe due to its simple structure, real-time visualization, rapid acquisition, and unique optical-slice characteristics. It has been widely used in the field of 3D measurements. A confocal microscope can determine the axial position of an object using a peak-search algorithm [2,3]. Compared with traditional microscopes, confocal microscopes can isolate nonfocal light outside the pinhole, known as the optical slice capability, allowing them to achieve submicron axial resolution. The intensity change in the optical slice is characterized by the energy change collected on the point detector at the imaging end. Overall, the use of free-form surfaces and reflection confocal microscopy provides significant advantages for achieving higher measurement accuracy in modern optical systems.

However, confocal microscopes have certain limitations. They are susceptible to power fluctuations from the light source, stray light from the environment, and common-mode noise in the measuring circuit. Additionally, the axial resolution of the confocal microscope can no longer satisfy the resolution requirement of the optical probe as the measurement accuracy of the freeform surface increases. To address these limitations, Wang et al. proposed an ideal noncontact probe structure in 2000, known as the differential confocal microscope [4], which has absolute measurement and focus-tracking advantages and improves the focusing sensitivity, sensor linearity, and signal-to-noise ratio. Most importantly, differential confocal microscopy (DCM) increases both axial and radial resolutions. They achieved a resolution of 2 nm within a measurement interval of 100 μm. In the subsequent two decades, numerous scholars have made considerable efforts to improve research on DCM and promote its development. For instance, Sun et al. proposed an axial high-resolution differential confocal microscope (AHDCM), which divides the detection optical path of a confocal system into three paths: the focal plane detection imaging optical, pre-focus detection imaging optical, and post-focus detection imaging optical paths [5]. The position information of the object to be measured was obtained by calculating the energy curves of the three optical paths along the axial scanning position. Yun et al. learned from the experience of Shepperd et al. [6] and extended it to the use of DCM, and they proposed a new method of DCM with D-shaped pupils (DDCM) for high-resolution 3D imaging of the surface microstructure of large-sized samples. Theoretical analysis and experimental results showed that the axial resolution of DDCM can reach 5 nm at a working distance of 3.1 mm, and the imaging speed can be increased by three times compared with that of a conventional confocal system at the same resolution [7]. Zou et al. introduced radial birefringent pupil filters into differential confocal systems and effectively improved the lateral resolution of these systems [8]. Additionally, Wang et al. proposed a high-precision differential confocal measurement method for filtering radially polarized light pupils, which improved the lateral resolution of the differential system by using radially polarized light and pupil filtering [9].

The popularization of differential confocal microscopy has enabled the transformation of theory into practical applications. Scholars have increasingly focused on addressing practical problems encountered in real-world scenarios. Two core issues have been identified: First, the measured object’s surface is often not a perfectly smooth plane perpendicular to the main optical path [10,11,12,13,14]. Second, there is a trade-off between measurement accuracy and single-point axial scanning efficiency when using DCM as the optical probe of the profiler [15,16]. To achieve higher axial resolution with DCM, the system must use a high numerical aperture (NA) objective [17] and improve its aperture utilization [18]. Reflective DCM requires the beam to be perpendicular to the test surface. However, test surfaces often have gradients, as illustrated in Figure 1. When the gradient exceeds a certain threshold, some of the energy is lost because it cannot return to the main optical path. Figure 1 shows the cause of the “lost” energy of the object and the image sides. To increase the NA utilization rate, the gradient threshold has been further reduced. This reduction has led to a decrease in the axial resolution of the DCM, which significantly increases the measurement uncertainty due to a decrease in energy and signal-to-noise ratio (SNR) (depending on the system parameters). Moreover, the slope of the test surface affects the optical aberration and artifacts of the microscopic objective and leads to a significant deviation between the measured and actual results [12,13]. Mauch et al. conducted a detailed analysis and verification of the focus shift phenomenon of a microscopic objective lens while measuring a free-form surface [12].

In our previous study, we proposed a slope measurement system structure and algorithm based on double cylindrical mirrors [14]. This system realized a differential probe that could simultaneously obtain the spatial position and slope of points on the tested surface. In addition, some scholars also focus their work on slope measurement [19,20,21], including Cacace’s 2009 research on slope measurements using position-sensitive detectors (PSD) [19]. Unfortunately, these studies failed to consider the increased spatial position measurement errors and decreased resolution caused by the larger slopes of the points being measured.

In addition to the slope of the measured object, the efficiency of laser differential confocal microscopy used for real-time measurements is also a major focus of research. This type of microscope uses a high-precision scanning mechanism, such as piezoelectric ceramics and voice coil motors, carrying a high NA objective for axial scanning of the surface under test. However, a small axial scanning interval inevitably leads to a decrease in the measurement efficiency of differential confocal microscopy. This creates a contradictory relationship between measurement accuracy and efficiency, with accuracy decreasing as the scanning step increases [15]. Moreover, the positioning error of the axial scanning displacement mechanism cannot satisfy uniform sampling, resulting in a large error in the linear fitting (LF) calculation at the zero-crossing. This issue further compromises the accuracy of the measurement. To improve the efficiency and accuracy of laser differential confocal microscopy, researchers must carefully consider the impact of the slope of the measured object and the positioning error of the scanning mechanism.

To address the aforementioned issues, we propose a new strategy for computing the Pearson correlation coefficient (PCC) based on the peak-clustering algorithm, which we apply to an optical probe system using DCM. The flowchart of our proposed PCC compensation strategy is shown in Figure 2. This strategy resolves the trade-off between measurement accuracy and efficiency in DCM scanning by enlarging the sampling interval and improving scanning efficiency while maintaining set accuracy. The scanning transmission mechanism is more tolerant of positioning errors, and the method compensates for the problem of the surface slope not being perfectly perpendicular to the optical axis. These improvements enhance the robustness of the DCM noncontact optical probe, improve the overall accuracy of 3D measurements, and reduce the uncertainty of the system. In this study, we present our new system structure and mathematical model, which combine the Debye integral and Fresnel integral to account for the NA of the microscopic objective and the image square focusing lenses. We introduce the PCC strategy in Section 3 and describe a simulation experiment in Section 4. Physical experiments and discussions are presented in Section 5, and we summarize our findings and draw conclusions in Section 6.

## 2. Numerical Model

Figure 3 depicts the DCM system structure, which utilizes a high-quality, monochromatic Gaussian beam generated by a HeNe laser with good coherence. The beam is split by the beam expander, and a plane wave is directed through the microscopic objective. The focal point of the beam extension can be treated as a good point source, and the parallel beam is focused on the surface of the object to be measured (SUT) via the high-NA objective. The parallel beam is then split into two beams by splitter B after passing through the focusing lens with a low NA, and it enters the corresponding detectors via pinholes A and B, which are located behind and in front of the focus, respectively. As the piezoelectric (PZT) moves the microscopic objective along the axis, the I_A_ and I_B_ curves are formed on the two detectors. Due to the symmetry of the two pinholes concerning the focal points, the energy curves of the two detectors should be symmetrical about the corresponding axial positions of the focal points [4]. The axial light intensity response varies based on the difference in reflectance of the tested surface, with lower reflectance leading to a lower peak value. We address this issue by using a normalized, anti-reflectance, differential confocal, axial light intensity response formula for data processing, which eliminates the influence of surface reflectance and enables precise focus determination of the profile of the measured surface with different reflectances. The mathematical model of the DCM typically starts with Fresnel diffraction and deduces a scalar mathematical model using Fourier transform.

The NA is a dimensionless quantity that describes the angular range of light collected by a lens and reflects the optical system’s ability to converge light into a beam. Optical systems with an NA greater than 0.7 are commonly referred to as strong focusing systems. When the NA is less than 0.7, the scalar diffraction theory is often used to theoretically derive the diffraction focusing problem, according to most scholars. However, common confocal microscopy imaging theories and techniques are based on paraxial approximation conditions, using an objective lens with a lower NA, and ignoring the polarization effect of the incident light of the objective lens. When the numerical aperture of the objective lens is 0.7, the impact of these effects on the imaging characteristics of the system becomes significant and must be considered in the imaging theory [22]. In addition, Tan et al. (2016) verified, based on theoretical and experimental exploration for confocal microscopy detection, that when the NA is higher than 0.4 [23], the change caused by polarization must be considered. This is why the microscope’s NA is marked as 0.4 in the picture showing the optical path structure. The existing literature shows that the NA should be ≥0.4 when using the Debye integral. Otherwise, the distribution of the axial point spread function (APSF) contains an error that increases with NA. Moreover, Tan proposed that an accurate theoretical calculation of the APSF should be performed. The Debye integral can be used to calculate the intensity distribution in nonparaxial imaging, replacing the original theoretical model based on Fresnel scalar diffraction. For the probe system structure shown in Figure 3, both a microscopic objective lens with a high NA and an image square focusing lens with a low NA were used. Furthermore, the influence of the surface gradient was considered, and a more practical mathematical model of the system was derived.

Figure 4 displays the unfolded model of a DCM. To simplify, pre-focal and post-focal light-field distributions represent the pre-focal and post-focal situations, respectively. This is due to the previously mentioned symmetry. The DCM optical paths can be divided into illuminating and detecting optical paths, which are separated by the sample under test reflection process.

In this case, a practical lens is considered circularly symmetrical, and the pupil function is only a function of the radial coordinates, that is, P(*x*, *y*) = P(*r*). Its pupil function is only a function of the radial coordinates, that is, P(*x*, *y*) = P(*r*). The pinholes used in the study were circular and symmetrical. Therefore, using the polar form in the formula derivation such as *U*_0_(*x*_0_, *y*_0_) is equivalent to *U*_0_(*r*_0_, *ψ*_0_). First, when the light field of the point source is known, the parallel light field *U*_1_(*x*_1_,*y*_1_) can be deduced using the Huygens–Fresnel diffraction integral formula
(1)$$U_{1}\left(r_{1}{,\psi }_{1}\right){=m}_{1}\times \int_{0}^{2\pi }\int_{0}^{\infty }U_{0}\left(r_{0}{,\psi }_{0}\right)\times \;\exp\left(-\frac{ik}{{2f}_{0}}\left(r_{0}{}^{2}{+r}_{1}{}^{2}{\;-\;2r}_{0}r_{1}\cos\left(\psi_{0}{-\;\psi }_{1}\right)\right)\right)r_{0}{dr}_{0}{d\psi }_{0},\;$$
where *m*_1_ is a constant coefficient, *k* is 2π/λ, and *r*_0_ and *ψ*_0_, respectively, represent the polar diameter and polar angle in the planar polar coordinate system of *U*_0_. When the distance between SUT and the focus of the microscopic objective is u, based on obtaining parallel light fields, *U*_2_(*r*_2_, *ψ*_2_, *u*) of the light field formed on SUT can be obtained based on the Debye diffraction integral formula and Hankel transformation [22,23],
(2)$$U_{2}\left(r_{2}{,\psi }_{2},u\right){=m}_{2}\;\times \;\frac{i}{\lambda }\iint_{\Omega }^{\;}P\left(\theta_{d}{,\varphi }_{2}{,U}_{1}\right)\exp\left[{-ikr}_{2}{\sin\theta }_{d}\cos\left(\varphi_{2}{\;-\;\psi }_{2}\right){\;-\;iku\cos\theta }_{d}\right]{\sin\theta }_{d}{d\theta }_{d}{d\varphi }_{2},$$
where *P* is the pupil function of the microscopic objective lens, *m*_2_ is the constant coefficient, and Ω is the integral region, *r*_1_ and *ψ*_1_, respectively, represent the polar diameter and polar angle in the planar polar coordinate system of *U*_1_. Additionally, $\theta_{d}$ represents the angle between the incident ray and the optical axis ($\theta_{d}$: 0 < ${\;\theta }_{d}<\theta_{d\max}$,${\;\theta }_{d\max}$ = arcsin (NA/*n*)), NA is the numerical aperture of the lens, *n* is the image square refractive index, *φ* is the circular angle. The gradient of the SUT exists in a two-dimensional (2D) form. The gradient was converted into polar coordinates as follows:*T*(*θ_x_*, *φ_y_*) = *T*(*α_t_*, *γ_t_*),(3)

As shown in Figure 5, the gradient was decomposed into the depth and direction of the tilt. At the energy detection angle, the tilt direction does not affect the amount of energy received by the image.

Simultaneously, because the SUTs obey the law of reflection, the optical axis of the returned beam changes by twice the depth of the gradient. Herein, we propose the simulation of the generation of a tilt by adding a phase to the SUT reflection process to change the wavefront phase. The beam reflected by the SUT in the probe path can be regarded as an ideal point source, and the Debye diffraction integral formula is used to simulate the propagation process. The parallel optical field *U*_3_(*r*_3_, *ψ*_3_, *u*) restored through the microscopic objective can be expressed as
(4)$$U_{3}\left(r_{3}{,\;\psi }_{3},\;u\right){=m}_{3}\;\times \;\frac{i}{\lambda }\iint_{\Omega }^{\;}P\left(\theta_{3}{,\;\varphi }_{3}{,\;U}_{2}\;\times \;\exp\left(ikr_{2}{\cos\psi }_{2}\tan2\alpha \right)\right)\times \exp\left[{-ikr}_{3}{\sin\theta }_{3}\cos\left(\varphi_{3}{\;-\;\psi }_{3}\right){-\;iku\cos\theta }_{3}\right]{\sin\theta }_{3}{d\theta }_{3}{d\varphi }_{3},$$
where α is the “depth” of the current gradient of SUT, and *m*_3_ is the constant coefficient. *U*_4_(*r*_4_, *ψ*_4_, *u*, −*u_m_*) and *U*_5_(*r*_5_, *ψ*_5_, *u*, +*u_m_*) are the distribution of the front and rear optical fields, respectively, obtained by Fresnel’s diffraction integral formula,
(5)$$U_{4}\left(r_{4}{,\;\psi }_{4}{,\;u,\;-u}_{m}\right){=m}_{4}\;\times \int_{0}^{2\pi }\int_{0}^{\infty }U_{3}\left(r_{3}{,\psi }_{3},u\right)\times \exp\left(-\frac{{ikr}_{3}{}^{2}}{{2f}_{2}}\right)\times \;\exp\left(-\frac{ik}{{2f}_{0}{-2u}_{m}}\left(r_{3}{}^{2}{+r}_{4}{}^{2}{\;-\;2r}_{3}r_{4}\cos\left(\psi_{3}{\;-\;\psi }_{4}\right)\right)\right)r_{3}{dr}_{3}{d\psi }_{3},$$
(6)$$U_{5}\left(r_{5}{,\;\psi }_{5}{,\;u,+u}_{m}\right){=m}_{5}\;\times \int_{0}^{2\pi }\int_{0}^{\infty }U_{3}\left(r_{3}{,\psi }_{3},u\right)\times \exp\left(-\frac{{ikr}_{3}{}^{2}}{{2f}_{2}}\right)\times \;\exp\left(-\frac{ik}{{2f}_{0}{+2u}_{m}}\left(r_{3}{}^{2}{+r}_{5}{}^{2}{\;-\;2r}_{3}r_{5}\cos\left(\psi_{3}{\;-\;\psi }_{5}\right)\right)\right)r_{3}{dr}_{3}{d\psi }_{3},$$

Among them, *m*_4_ and *m*_5_ are constant coefficients. The signal intensities from detectors B and A can be expressed by Formulas (7) and (8), respectively, and the normalized focus error signal (FES) can be expressed by Formula (9).
(7)$$I_{B}=\iint_{D_{f}}^{\;}U_{4}\left(r_{4}{,\;\psi }_{4}{,\;u,\;-u}_{m}\right)r_{4}{dr}_{4}{d\psi }_{4},$$
(8)$$I_{A}=\iint_{D_{b}}^{\;}U_{5}\left(r_{5}{,\;\psi }_{5}{,\;u,\;-u}_{m}\right)r_{5}{dr}_{5}{d\psi }_{5},$$
(9)$$\bar{FES}=\frac{I_{B}{-\;I}_{A}}{I_{B}{+I}_{A}},$$

Normalization aids in the elimination of common-mode noise, especially for changes in the reflectivity of the measured object with great robustness, as shown in Figure 3. In addition, it has a strong inhibitory effect on changes in the laser power, transmission noise in the optical path, and responsiveness of the photodiode.

## 3. PCC Strategy

To solve the problem of the presence of a gradient, a large step length, and nonuniform sampling on the SUT surface affecting the detection accuracy of the optical probe position in a DCM and obtaining the slope of the measured point, we proposed a PCC strategy.

Figure 6 shows the mathematical model constructed above. When the micro-objective is driven by the micro-objective driver with a small step length and negligible positioning error, and by the linear motor with a long stride and a large amount of shaking, the data are obtained from photodetector A, namely, the normalized energy relation curve with axial displacement. Red represents ideal uniform data with a small step length, step size D_s_, and negligible positioning error. The blue data are the data with large and uneven positioning errors, as shown in Figure 6. D_sc1_ and D_sc2_ are the sampling intervals when the positioning accuracy of the transmission mechanism is insufficient. At the same signal-to-noise ratio, the zero-crossing position obtained using the blue data yielded a large error. The PCC is a common similarity measure that can handle multidimensional data clustering classes. The minimum PCC value was −1 (opposite vector), and the maximum PCC value was +1 (same vector). Owing to its excellent recognition and matching abilities, we hope to use this algorithm effectively to solve the existing problems in DCM systems.

### 3.1. Data Processing Overview

The process for the accurate compensation of the zero-crossing position of the differential confocal response signal was mainly divided into two parts: axial position homogenization based on virtual interpolation and rapid matching recognition based on peak clustering.

We define the distance from the nearest sampling point to the zero-crossing position in the forward direction (FES > 0) of the uniform differential confocal response curve as the zero-crossing sampling offset (ZCSO). In the calibration experiment, a small and uniform step size was used to drive the objective lens driver to conduct axial scanning on fixed measured points, and the differential confocal response signal data were recorded at different slope intensities. The overall data were then grouped into different ZCSOs according to large and uniform step sizes.

For real-time acquisition, the major-step response of the signal differential confocal raw data, when it passes the zero crossing, depends on the symmetry principle, which is the minimum value in the positive and negative directions of the data capture (*N* − 1)/2 (with zero tilt when the energy peak position of the two detectors is the boundary).

First, the interception of the *N* data was conducted in the axial location based on virtual interpolation homogenization, achieving an equidistant differential confocal response signal. Accurate compensation was then realized using the fast-matching recognition algorithm, based on the peak clusters of the calibration experiment performed for different slope strengths and the ZCSO using the PCC matching of the data access to the zero-crossing position—the accurate position.

### 3.2. Axial Position Homogenization Based on Virtual Interpolation

Calculation of the PCC requires the same vector dimensions and position matching. Simultaneously, it was assumed that the sloshing amount of the standard data could be ignored as isometric data. Therefore, it is necessary to preprocess the measured data obtained in real-time. The generalized expressions for these axial scan positions at unequal intervals are as follows:(10)$$\vec{D}=[D_{1},\;D_{2},\;D_{3}\;,\ldots ,\;D_{N-1},\;D_{N}]$$
where *N* is the number of data points obtained by a single axial scan, that is, the vector length. We define the generalized expression of virtual interpolation as
(11)$${\vec{D}}_{t}=[D_{1}+\varepsilon ,\;D_{1}+\varepsilon +1\;\times \;\Delta d,\;D_{1}+\varepsilon +2\;\times \;\Delta d,\ldots ,\;D_{1}+\varepsilon +(N\;-\;2)\;\times \;\Delta d,\;D_{1}+\varepsilon +(N\;-\;1)\;\times \;\Delta d]$$
where ε is the defined virtual interpolation bias, and Δd is the axial sampling interval when the motor does not have a positioning slosh error set by the system. The selection of ε determines the accuracy of the homogenization algorithm and then affects the accuracy of the entire matching strategy. The selection principle of ε is to make ${\vec{D}}_{t}$ as close to non-equidistant actual data as possible; thus, we defined the following objective function:
(12)$$f(\varepsilon )=\mathrm{argmin}(\sum_{i=1}^{N}{\left(\min\left(\mathrm{abs}\left(D_{1}+\varepsilon +(i\;-\;1)\;\times \;\Delta d\right]\right)-[D_{1},\;D_{2},\;D_{3}\;,\ldots ,\;D_{N-1},\;D_{N}])\right)}^{2}$$

As the positioning accuracy of the calibration data is far less than that of the sampling interval expected by the system, which is Δd, the following approximation can be obtained,
(13)$$\min\left(\mathrm{abs}\left(\left(D_{i}+\varepsilon +(i\;-\;1)\;\times \;\Delta d\right]\right)-[D_{1},\;D_{2},\;D_{3}\;,\ldots ,\;D_{N-1},\;D_{N}\right]))\approx \left|D_{1}+\varepsilon +(i\;-\;1)\;\times \;\Delta d\;-\;D_{i}\right|$$

Based on this approximation, the partial derivative of f (ε) is calculated and set to zero. Because the function has an obvious and unique minimum point, it can be solved as follows:(14)$$\frac{\partial (f\left(\varepsilon \right))}{\partial \varepsilon }=\frac{\partial (\sum_{i=1}^{N}{\left(\left|D_{1}+\varepsilon +(i\;-\;1)\;\times \;\Delta d\;-\;D_{i}\right|\right)}^{2})}{\partial \varepsilon }=2\;\times \overset{N}{\underset{i=1}{\sum }}(D_{1}+\varepsilon +(i\;-\;1)\;\times \;\Delta d\;-\;D_{i})$$

Based on the above formula, we can obtain
(15)$$\varepsilon =\frac{-\sum_{i=1}^{N}(D_{1}+\varepsilon +(i\;-1)\;\times \;\Delta d\;-\;D_{i})}{N}$$

The intensity sequence $I_{{\vec{D}}_{t}\left(i\right)}$ of the uniform data was then obtained based on linear interpolation.

### 3.3. Rapid Matching Based on Peak Clustering

We aim to address the issue of uncertainty in the axial position measurement of the differential confocal microscope by matching the similarity between the collected (actual) and calibrated data. In signal processing, measuring the degree of correlation between signals is often necessary for the statistical purposes of random variables or to determine the degree of correlation between signals. The correlation coefficient is a widely used quantitative index of the strength of statistical relationships in various fields, such as data science, digital image processing, matching, and biomedical signal processing. The Pearson product-moment correlation coefficient (PCC), also known as the Pearson correlation coefficient, is extensively used due to its complete theoretical proof and simple application, and it was proposed by Karl Pearson [24]. It measures the correlation between two variables and is defined as the quotient of the covariance and standard deviation between the two variables. The overall correlation coefficient can be expressed as:(16)$${\mathrm{Pearson}(I}_{{\vec{D}}_{t}}{,\;I}_{{\vec{D}}_{p,q}})=\frac{{\mathrm{Cov}(I}_{{\vec{D}}_{t}}{,\;I}_{{\vec{D}}_{p,q}})}{\sqrt{{V(I}_{{\vec{D}}_{t}})}\;\times \sqrt{{V(\;I}_{{\vec{D}}_{p,q}})}}=\frac{{E[(I}_{{\vec{D}}_{t}}\;\;{-\;\mu }_{I_{{\vec{D}}_{t}}}{)(\;I}_{{\vec{D}}_{p,q}\;}{-\;\mu }_{{\;I}_{{\;D\;\to }_{p,q}}})]}{\sqrt{{V(I}_{{\vec{D}}_{t}})}\;\times \sqrt{{V(\;I}_{{\vec{D}}_{p,q}})}}$$
where Cov is the covariance, $\sqrt{V}$ is the standard deviation, and E is the expectation. The Pearson correlation coefficient for individual samples can be obtained through the following formula [25]:(17)$$\mathrm{Pearson}\left(I_{{\vec{D}}_{t}},\;I_{{\vec{D}}_{p,q}}\right)=\frac{\frac{1}{N}\sum_{i=1}^{N}\left(I_{{\vec{D}}_{t}}\left(i\right)-\bar{I_{{\vec{D}}_{t}}}\right)\times \left(\;I_{{\vec{D}}_{p,q}}\left(i\right)-\bar{\;I_{{\vec{D}}_{p,q}}}\right)}{\sqrt{\frac{1}{N}\sum_{i=1}^{N}{\left(I_{{\vec{D}}_{t}}\left(i\right)-\bar{I_{{\vec{D}}_{t}}}\right)}^{2}}\times \sqrt{\frac{1}{N}\sum_{i=1}^{N}{\left(\;I_{{\vec{D}}_{p,q}}\left(i\right)-\bar{\;I_{{\vec{D}}_{p,q}}}\right)}^{2}}}$$

To obtain similarity for compensation, the enumeration method is commonly used to calculate each group of measured and calibrated data. However, the calibration dataset is usually large, and the enumeration method may result in a long calculation time, which is unsuitable for real-time compensation systems. To address this, we propose a PCC calculation method based on peak clustering. The proposed method is based on the concept that the clustering vector always points in the direction of increasing probability density [26,27], where the closer the displacement and tilt angle of the calibration data are to the experimental data, the higher the PCC will be. The peak position of the PCC can be extracted using the shortest path to avoid wasting data processing speed due to invalid data operations. The proposed algorithm is presented in Algorithm 1.
**Algorithm 1** A fast Pearson correlation coefficient calculation method based on Meanshift **Input:** Calibration data set: ${\;I}_{{\vec{D}}_{p,q}}$, Interpolated experimental data: $I_{{\vec{D}}_{t}},$   Iterative convergence threshold: ε   Minimum spatial scale: $h_{\min}$ **Output:** The horizontal and vertical coordinates corresponding to the positions with the highest similarity in the calibration data $\left\{P_{h}{(n),\;P}_{v}\left(n\right)\right\}$ 1: Initialize $P_{h}\left(n\right)$←m/2,$P_{v}\left(n\right)$←n/2,n←0, search space radius h←min(m, n)/2 2: Repeat 3: k←k + 1 4: Create a circular Gaussian mask $G_{m\times n}$$\;\mathrm{with}\;\left\{P_{h}{(k),\;P}_{v}\left(k\right)\right\}$ as the left and *h* as the radius, and set the mask values in regions beyond the radius *h* to zero 5:$P_{h}\left(k\right)\leftarrow \frac{\sum_{i=1}^{m}\sum_{j=1}^{n}{i\;\times \;\mathrm{Pearson}(I}_{{\vec{D}}_{t}}{,\;I}_{{\vec{D}}_{p,q}}{)\;\times \;G}_{m\times n}\left(i,\;j\right)}{\sum_{i=1}^{m}\sum_{j=1}^{n}{\mathrm{Pearson}(I}_{{\vec{D}}_{t}}{,\;I}_{{\vec{D}}_{p,q}}{)\;\times \;G}_{m\times n}\left(i,\;j\right)}$, $P_{h}\left(k\right)\leftarrow \frac{\sum_{i=1}^{m}\sum_{j=1}^{n}{j\;\times \;\mathrm{Pearson}(I}_{{\vec{D}}_{t}}{,\;I}_{{\vec{D}}_{p,q}}{)\;\times \;G}_{m\times n}\left(i,\;j\right)}{\sum_{i=1}^{m}\sum_{j=1}^{n}{\mathrm{Pearson}(I}_{{\vec{D}}_{t}}{,\;I}_{{\vec{D}}_{p,q}}{)\;\times \;G}_{m\times n}\left(i,\;j\right)}$ 6: *h*←$h_{min}+\left(1-\frac{2h_{min}}{min\left(m,n\right)}\right)\times \sqrt{{\left(P_{h}\left(k\right)-P_{h}\left(k-1\right)\right)}^{2}+{\left(P_{v}\left(k\right)-P_{v}\left(k-1\right)\right)}^{2}}$ 7: Until convergence: $\sqrt{{{(P}_{h}{(k)-P}_{h}(k-1))}^{2}{+(P}_{v}{(k)\;-\;P}_{v}{\left(k-1)\right)}^{2}}\leq \varepsilon$ 8: $P_{h}\left(k\right)\leftarrow {\mathrm{round}(P}_{h}{(k)),\;P}_{v}\left(k\right)\leftarrow {\mathrm{round}(P}_{v}\left(k)\right)$


For the kernel function used in the clustering algorithm, the classical Gaussian kernel function was selected, and the corresponding weights were assigned to different positions in the computing space:(18)$$G_{m\times n}\left(i,\;j\right)=\left\{\begin{array}{c} \frac{1}{\sqrt{2\pi }h}\exp\left(\frac{-\left({\left({i\;-\;P}_{h}\left(k\right)\right)}^{2}+{\left({j\;-\;P}_{v}\left(k\right)\right)}^{2}\right)}{{2h}^{2}}\right),\;{\left({\;i\;-\;P}_{h}\left(k\right)\right)}^{2}+{\left({j\;-\;P}_{v}\left(k\right)\right)}^{2}{<\;h}^{2}\\ \;0\;,\mathrm{else} \end{array}\right.$$

Using this method, the zero-crossing position corresponding to ${\;I}_{{\vec{D}}_{P_{h}\left(k\right),P_{v}\left(k\right)}}$ of the calibration data with the highest PCC can be obtained, that is, the measured value of the current front axial position after compensation. Using the number *N*/2 sampling points of ${\;I}_{{\vec{D}}_{P_{h}\left(k\right),P_{v}\left(k\right)}}$, the set of calibration data with the highest similarity, and their corresponding small sampling interval difference signal zero-crossing distance, the number *N*/2 sampling points of the measured data $I_{{\vec{D}}_{t}}$, and distance from its zero-crossing distance can be obtained. The method that uses Meanshift to extract the peak PCC avoids traversing the entire calibration dataset and only needs to calculate the PCC of the calibration data contained in the sliding window space in the Meanshift calculation process, thus greatly reducing the number of required computations. This method not only improves the speed of data processing and satisfies the real-time performance of the system, but also improves the measurement accuracy of the system.

## 4. Simulations

In the second part, we consider the impact of high NA on the microscopic objective lens used. The signal output responses at the two pinholes were deduced by combining the Debye and Fresnel diffraction integrals, neglecting aberration effects, based on relevant published studies. Simulations were then conducted to validate the proposed algorithm using the mathematical model developed above. Specifically, a monochromatic laser with a wavelength of 642 nm was used as the light source in the simulations, and a collimating lens with NA = 0.2 was used to produce a parallel Gaussian beam with a diameter of 6.4 mm. The microscope had a focal length of 2 mm, NA = 0.7, and an image square lens focal length of 0.2, with a focal length of 100 mm. The angular depth in the standard dataset was 0.1° with a step length of 0−15°, and the axial scanning step was 10 nm. The simulation physical experiment used a scanning step of 1 μm (±0.1), and a random matrix was generated to simulate errors caused by motor positioning slosh during measurement.

The normalized energy-response curves of the detector behind the pinhole with a defocusing distance of 650 μm behind the focus at different tilt angles are shown in Figure 7. The curves represent six different dip depths in the range of 0–10°. Figure 7 shows that as the tilt angle increases, the peak energy value decreases gradually, but the axial position corresponding to the peak position remains unchanged. It is important to note that the variation in each energy curve at different tilting depths was not obtained based on the response curve at time zero, which was reduced by the same multiple along with the axial position.

The sensitivity of the DCM system, which determines its resolution and accuracy, is represented by the slope of the linear interval near the zero-crossing of the FES curve. Figure 7 shows that as the slope depth increases, the slope at the zero-crossing decreases, resulting in a considerable decrease in resolution and accuracy. The slope continued to decline but the change was not significant between 0–6°, while at 8°, the slope changed significantly. The slope decreased as the angle increased, but the change was not linear.

The normalization operation of the FES eliminates the influence of the SUT reflectance and reduces the effect of common-mode noise and light source fluctuations on the axial position measurement of differential confocal microscopy. However, simulation experiments (as shown in Figure 7 and Figure 8) have demonstrated that normalization alone cannot eliminate the effect of the decline in the zero-crossing slope. At large tilt depths, the DCM system’s accuracy must be corrected, as this can impact the measurement results of the surface being measured. It is worth noting that the zero-crossing slopes of the FES curves with different system parameters vary significantly at each dip depth, with the NA and focal length of the microscope objective being the primary factors affecting this.

Figure 9a,b displays the change process trajectory of the data array calculated by the PCC strategy and the peak clustering vector when ZCSO is 0.5 μm and the tilt angle is 5°, and when ZCSO is 0.3 μm and the tilt angle is 6°. When the enumeration method is used for continuous surfaces, the current data and all the datasets of tilt depth and ZCSO are used as the fitting surface for the similarity matching results. The blue-covered area is the dataset lattice calculated using the PCC strategy, the purple square is the core coordinate of the peak clustering vector during the calculation, and the arrow direction indicates the direction of the calculation path change. It is important to note that the angle between the two boundaries farthest from the center is chosen as the initial iteration point to demonstrate the algorithm’s effectiveness, while typically the starting iteration point is in the central region. Figure 9 clearly illustrates that the PCC strategy significantly reduces the calculation of similarity processing, increases the processing speed, and guarantees the system’s real-time target.

The PCC error distribution was obtained through simulation experiments at all tilting intensities and ZCSO conditions, as shown in Figure 10. The central lattice’s color represents the error level, with one dimension remaining unchanged while the error curve changes with the other shaft. The error level decreases and then increases as the tilt strength changes, except near the zero-crossing, where the error level first decreases and then increases. After multiple variable-parameter simulations, it was concluded that the error turning into a larger tilt strength was related to the system parameters, but the overall trend remained the same.

To further demonstrate the effectiveness of the proposed algorithm, we conducted simulation experiments. Firstly, we constructed a virtual 2D curve, represented by the solid green line in Figure 11a. The curve was composed of line segments at various oblique angles, with a radial length of 20 μm and a high-axial vector of 0.48 μm. The virtual probe continuously performed axial scanning of the measured points along the axis. To ensure high confidence, we conducted 1000 simulations at each given measurement point position. The system parameters were kept constant using a probe. The sampling steps of the calibration dataset and simulation physical experiment were consistent with those mentioned earlier, namely, 10 nm and 1 μm (±0.1), respectively, and the radial sampling interval was 0.4 μm. To simulate the noise effect in the physical experiment, we set the signal-to-noise ratio to 40 dB, and a random ZCSO was also set. The blue area in the figure represents the range of the measured point positions obtained by the PCC during the simulation’s physical experiment of the current signal-to-noise ratio, and the green area represents the range of positions obtained by traditional linear fitting. The figure indicates that the proposed compensation algorithm can significantly reduce the measured error range at each measured position. Figure 11b illustrates the error distribution when the tilt intensity is 10°, where the horizontal axis represents the axial measurement position error, and the vertical axis represents the distribution number of 1000 repeated measurement points at different error values. It can be observed that the errors are distributed based on the Gaussian distribution, the error distribution of the compensation algorithm proposed in this study is more concentrated, and the number of points falling at zero is significantly greater than that of the traditional zero-crossing fitting algorithm. Moreover, the zero-crossing fitting method inherently contains errors. Therefore, the proposed compensation strategy is proven to be effective in reducing the original error range and enhancing the system’s robustness.

In Figure 12, we selected four angles within the measurement range and a random ZCSO to track a single point 2000 times repeatedly to obtain the absolute value of the PCC compensation algorithm error (PCC_AME_), the absolute value of the linear fitting error (LF_AME_), the PCC compensating algorithm standard deviation (PCC_SD_), and the linear fitting standard deviation (LF_SD_). The bar chart demonstrates that the error generated by the algorithm proposed in this study is significantly smaller than the error generated by linear fitting, regardless of the absolute mean value of the error or standard deviation.

## 5. Experimental Results and Discussion

To verify the effectiveness of the compensation strategy proposed in this study and the PCC strategy, we constructed a noncontact optical probe structure for a differential confocal sensor, as shown in Figure 13. A single-wavelength fiber laser (LP642PF20, 642 nm, 20 mW, Thorlabs, Newton, NJ, USA) was used as the light source. To ensure imaging quality, the probe system used a squirrel-cage structure. First, a collimator (F810FC-635, NA = 0.25, f = 35.41 mm) was used to generate a Gaussian parallel optical field with a beam waist radius of 3.2 mm at the exit of the pupil’s surface. The collimated beam passed through the beam splitter, reached the microscopic objective lens (LMPLFLN 20×, NA = 0.4, f = 9 mm; Olympus, Tokyo, Japan), and was focused on the measured object.

After the parallel beam was focused using a plano-convex lens (LA1207-A, Ø1/2”, f= 100.0 mm, Thorlabs, Newton, NJ, USA), it was divided into pre- and post-focus measurement beams using a beam splitter. A pinhole probe (SMO5PD1B, O = 20 μm, Thorlabs, Newton, NJ, USA) with a defocus of ±650 μm was shot.

### 5.1. Standard Data Acquisition and Verification Experiment

The calibration experiment’s data acquisition was carried out using an ultrahigh-precision objective positioner and a high-precision six-axis displacement platform, which were responsible for axial scanning and angular tilt, respectively. The noncontact probe structure of the DCM was kept fixed, and a smooth aluminum sheet with 92% reflectivity was used as the standard subject. As depicted in Figure 12, the microscopic objective was controlled by the objective driver (P72.Z100S & E53.d, Harbin Harbin Core Tomorrow Science & Technology, Harbin, Heilongjiang, China) and was used for axial scanning. The objective lens driver’s axial resolution was less than 1 nm, and it can cycle and reciprocate in the range of 100 μm at a frequency of 10 Hz, driven by sinusoidal or triangular waves. This meets the system’s required response frequency and the required step size of ZCSO in standard datasets. The six-axis displacement platform (H-811. I2, ±10, Physik Instrumente, Karlsruhe, Germany) was used to construct the plane mirror. The platform’s rotation and repeated positioning accuracy were 2.5 urad, making it an ideal tool for tilt angle calibration.

The axial scanning compensation was 10 nm and the scanning range was 80 μm (<100 μm). The six-axis displacement platform controlled the plane mirror, achieving a step size of 0.25° and a range of 0–10°. ZCSO offset data were constructed for each group of angles using 10 nm as the step size and 0–1 μm as the offset range to obtain the standard dataset. To test the effectiveness of the compensation strategy proposed in this study and the efficiency of the proposed algorithm, 1000 sets of data were randomly selected as the test set for analysis to avoid contingencies. The test set was generated using MATLAB (version 2020a, MathWorks, Natick, MA, USA) by analyzing the tilt angle in the range of 0–10° and generating 1000 random 2D coordinates with offsets in the range of 0–1 μm as the tilt intensity and ZCSO. Random soil slosh amounts (with values ≤ 0.2 μm) were added to 80 uniformly selected sampling locations in the axial scanning range of 10–90 μm to represent the actual slosh amounts of the linear motor. The rotation center of the six-axis displacement platform was determined by the imaging device, and the center of the working interval of the objective positioner was within the focal depth range of the microscope. Finally, 1000 random data points were generated as the initial state of scanning for data acquisition. Figure 14 shows that the compensation strategy significantly improved accuracy in the entire test set, especially as the tilt angle increased. In the global range, the average error of the Pearson similarity coefficient compensation strategy was less than 10 nm, while the error of the traditional linear fitting was 25.7 nm. It is worth noting that acquiring annotated datasets requires maintaining good environmental conditions, such as vibration isolation and minimizing stray light, to maintain a high signal-to-noise ratio. If the environmental conditions are not ideal, the accuracy compensation effect of the compensation strategy will be reduced.

To verify the effectiveness of the rapid recognition algorithm based on the peak clustering algorithm, a personal computer equipped with the ThinkBook 14 G4+ IAP and the 12th Gen Intel(R) Core(TM) i5-12500H 2.50 was used. Figure 15a,b shows the computing efficiencies of the traversal method and PCC-based algorithm at different tilt intensities and offsets, respectively. It can be observed that the PCC algorithm has obvious efficiency advantages and satisfies the real-time requirements of the system. Over the entire range, the average processing time of the PCC algorithm was 0.02606 s, and the average time of the traversal method was 0.15677 s; this improved the efficiency by 83.37%.

### 5.2. Repeatability Experiment and Antidisturbance Test

The constructed differential focal noncontact probe was fixed on the Z axis of the precision 3D motion platform, as shown in Figure 16. To ensure accuracy, the feedback component of the Z axis used a grating developed at the National Grating Manufacturing and Application Engineering Technology Research Center in China. The maximum feedback accuracy was 0.01 μm, and the positioning slope of the Z-axis motor was ±0.2 μm when subjected to loading. The positioning sloshing of the XY axis was 0.1 μm. The measured object was placed on the electric 2D rotating yaw motion platform to ensure that the initial tilt bias did not affect the experimental results. The Z-axis motor was used to drive the DCM with a step size of 1 μm for axial scanning of standard objects.

To verify the repeated accuracy of the compensation strategy proposed in this study and to avoid contingencies, 100 repeated measurements were conducted at each angle, and the results were compared with those of the traditional LF method. The results are shown in Figure 17a,b, where the upper half of Figure 17a shows the statistical diagram of the probability distribution of errors, and Figure 17b shows the error distribution pairs. The accuracy of the compensation strategy was higher than that of LF, the error distribution under the compensation strategy was more concentrated, and the bias phenomenon of the error was less than that of LF. In addition, it can be observed that the distribution and error level of LF tended to increase as a function of the tilt angle.

In addition to the repeated-measurement verification experiments, we conducted an anti-interference experiment to verify the anti-interference capability of the compensation strategy. The Shenzhen direct current reduction motor (JGA25-370, Shenzhen, China) was the power machine with adjustable interference noise used to simulate the disturbances to which the optical probe may be subjected under working conditions. The experimental results are listed in Table 1. Table 1 shows that when the SNR gradually decreases, the traditional LF method produces increasingly larger errors, while the use of the compensation strategy reduces the error growth trend, which significantly improves the robustness of the system.

### 5.3. Grating Measurement Experiment

A large-grating engraving mechanism was used with quartz glass as the base and aluminum as the film layer (92% reflectivity). The carving knife was positioned on the surface of the film layer three times at different inclination angles; the waste material was not removed, and the degrees of adjustment of the carving knife were 3°, 6°, and 9°. The slot width ensured that the maximum spot size accommodated the defocusing range required to form the DCM response curve. The atomic force microscope (Dimension Icon, Bruker Corporation, MA, USA) has an axial resolution of 0.25 nm in Figure 18a; thus, its measurements satisfy our requirement to use it as a truth value. The measured results are shown in Figure 18b (the fitting results of only one grating are shown here for a condensed expression). In addition, seven sampling data positions on a random line on the three rasters were selected for measurement, registration, fitting, and analysis. The measurement results for the six angles are shown in Figure 19. The measurement error in the compensation strategy was significantly smaller than that of the traditional LF method. The average error at the compensation strategy was 17.537 nm, whereas the traditional LF error was 32.449 nm, which was slightly higher than the axial prediction error during calibration. This was because the insufficient positioning accuracy in the horizontal direction of the three-coordinate displacement platform resulted in an error in the point cloud registration, which affected the final measurement results.

### 5.4. Discussion and Error Analysis

In this study, we verified the advantages of the PCC compensation strategy over traditional LF and demonstrated the significance of the compensation strategy based on simulations and physical experiments in the case of large step lengths. Our length experimental results showed that the compensation strategy effectively corrects the bias problem of LF measurement points caused by long-step sampling and reduces system noise caused by increased angles and a decreased signal-to-noise ratio. The PCC compensation strategy also exhibited good robustness. Furthermore, the proposed PCC algorithm satisfies the real-time requirements of the system, providing feedback on the position of the measured points within periods that are less than half the time required for a common high-speed measurement cycle. Unlike traditional LF, which requires the determination of several sampling points near the zero-crossing position, the PCC compensation strategy effectively suppresses the bias problem of zero-crossing at large step lengths. However, it should be noted that our experiments were conducted under highly controlled environmental conditions with a good signal-to-noise ratio, thanks to the squirrel-cage structure and air-floating platform. When the signal-to-noise ratio decreases and the tilt angle increases, the effectiveness and significance of the PCC compensation strategy will be highlighted further. However, there are some limitations in this study, which will be solved gradually in future research. Our physical experiments used a microscopic objective lens with an NA of 0.4, which has good tolerance for angular tilt. However, if an objective lens with a larger NA is used, the LF error increases considerably. Specifically,

(1)It should also be noted that our experiments and algorithms only addressed the surface gradient and sampling problems for actual measurement conditions but ignored the impact of surface curvature [12]. However, in actual working conditions, the tested free surface may exhibit a phenomenon where the local curvature is too large to be regarded as an ideal place in the airy spot range. This introduces additional requirements for the speed and accuracy of similarity matching, which will be studied further in the future.(2)The accuracy of the compensation strategy is directly influenced by the establishment of a standard dataset. In this study, the sampling step for the physical experiments was 10 nm, which was chosen to satisfy the real-time requirements of the system. However, this step length introduces inherent errors of several nanometers in the axial position measurement, which is a significant source of error. To improve the compensation accuracy, it may be necessary to use a smaller scan step or another calibration method.(3)The implementation of the compensation strategy requires precise installation and adjustment of the axial scanning mechanism. Specifically, the direction of movement of the scanning mechanism should be perpendicular to the initial plane to minimize measurement errors. Moreover, the inconsistency between the Z-axis movement direction used in this study and the normal direction of the datum plane, as well as the misalignment of the system construction, can also contribute to measurement inaccuracies [28].(4)The data homogenization preprocessing methods proposed in this study to address the axial scanning slosh can be further optimized to improve their effectiveness.

## 6. Conclusions

In conclusion, differential confocal optical probes are a promising option for noncontact 3D measurements of free-form surfaces. However, the low measurement speed of point-by-point axial scanning poses a significant limitation. Additionally, traditional zero-crossing methods face inherent errors when dealing with large step lengths and surface slopes. To address these issues, we performed the following:(1)We introduced a compensation strategy for zero-crossing position detection based on the PCC, effectively reducing the error growth and enhancing the system’s robustness.(2)To ensure real-time performance, we proposed a fast similarity-matching algorithm based on peak clustering that achieves accurate position compensation within half a cycle of high-speed scanning.(3)We validated the proposed method through simulations and physical experiments that demonstrated good repeatability and disturbance resistance. With an NA of 0.4, the method achieved a slope repetition accuracy 10 nm better than that for slopes less than 12°.

Our approach is simple and efficient and has the potential to enable high-speed measurements of free-form surfaces with adequate accuracy. Further research can explore its practical applications in various industries, such as optical research, industry, and commercial fields.

## Figures and Tables

**Figure 1 micromachines-14-01163-f001:**
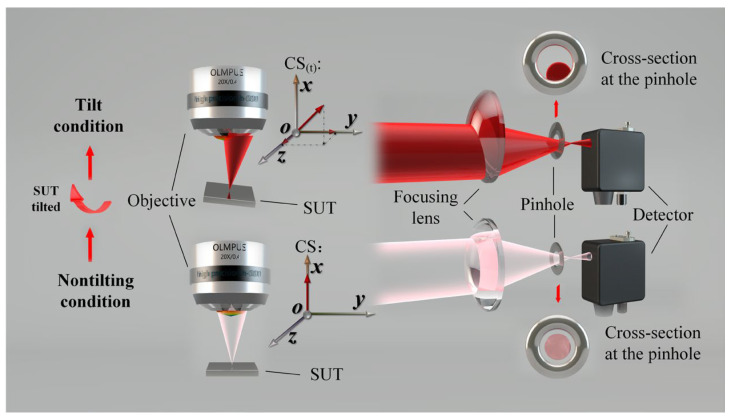
Analysis diagram of the cause of energy loss.

**Figure 2 micromachines-14-01163-f002:**
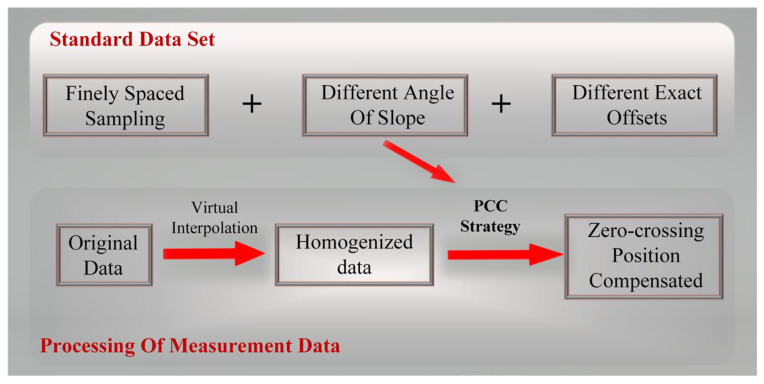
Flowchart of the PCC compensation strategy.

**Figure 3 micromachines-14-01163-f003:**
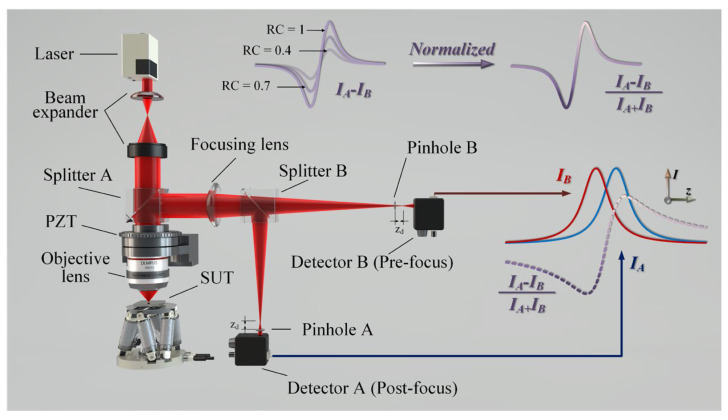
System architecture of the differential confocal microscope (DCM).

**Figure 4 micromachines-14-01163-f004:**
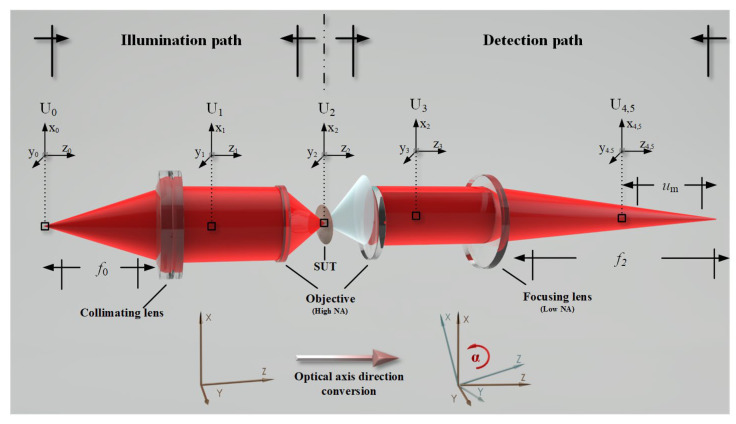
Expansion model of the DCM.

**Figure 5 micromachines-14-01163-f005:**
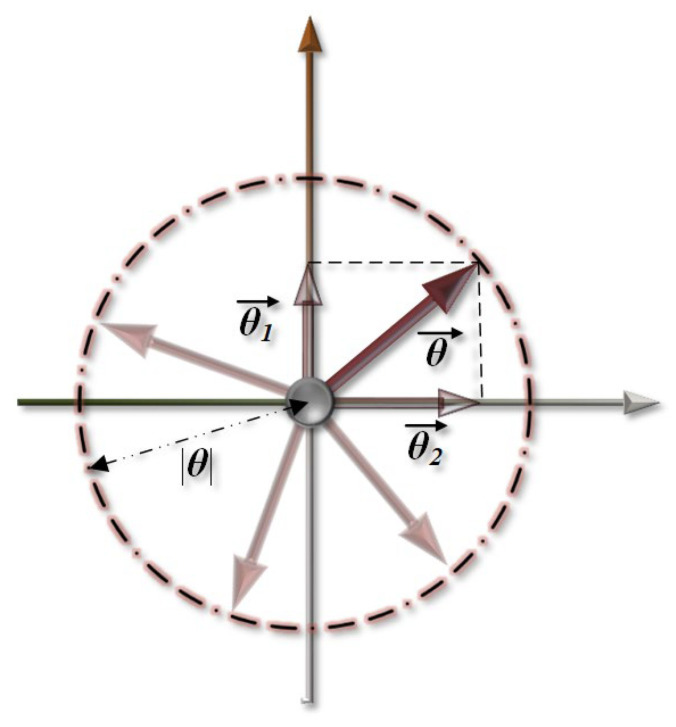
Diagram depicting the relationship between slope vector and depth.

**Figure 6 micromachines-14-01163-f006:**
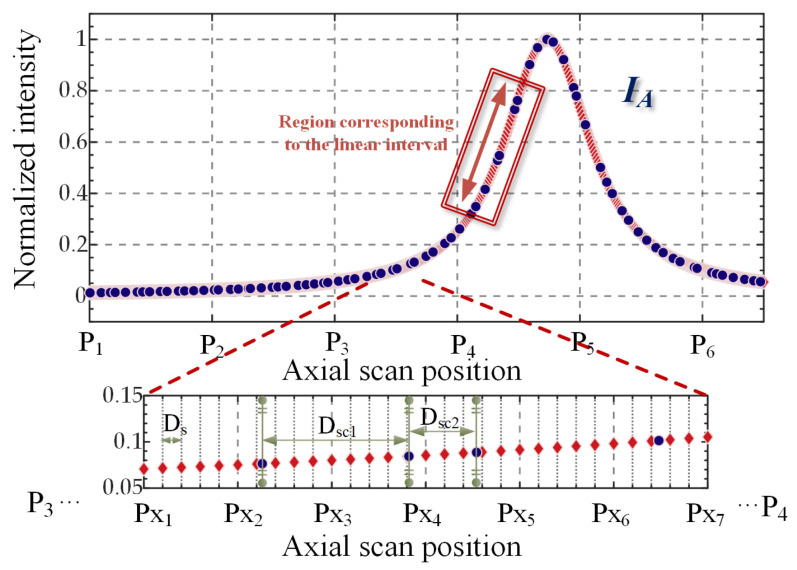
Unimodal data in the DCM.

**Figure 7 micromachines-14-01163-f007:**
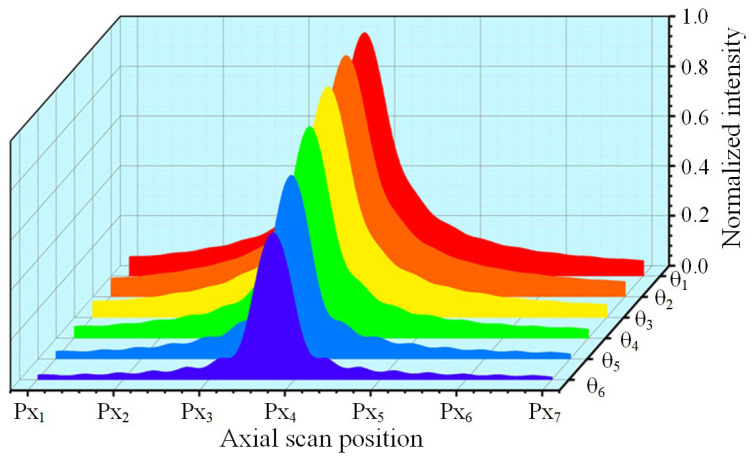
Energy response curve of the detector located behind the post-focal pinhole when the SUT is tilted at depths in the range of 0–10° (red to blue in steps of 2°).

**Figure 8 micromachines-14-01163-f008:**
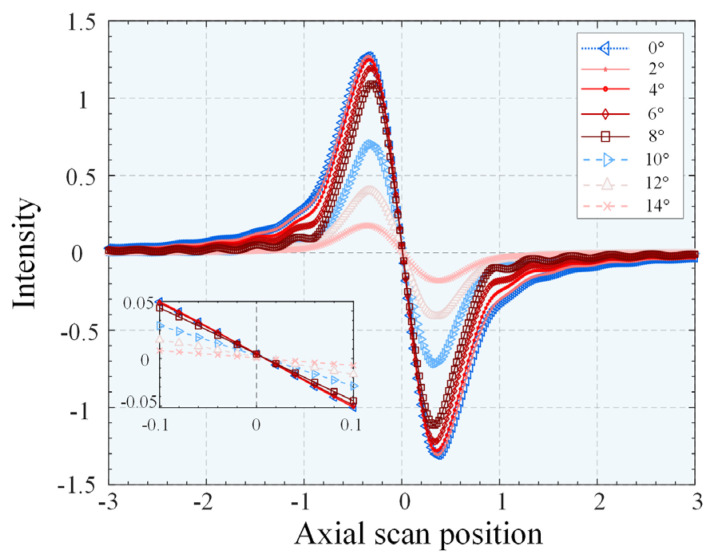
Focus error signal (FES) response curves of the differential confocal microscopy system at different tilt depths.

**Figure 9 micromachines-14-01163-f009:**
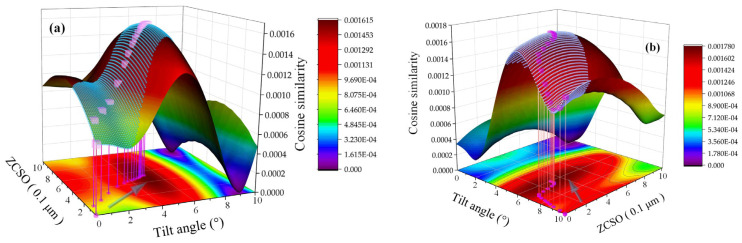
Comparison of calculation processes between the Pearson correlation coefficient (PCC) strategy and enumeration method: (**a**) 0.5 μm, 5°; (**b**) 0.3 μm, 6°.

**Figure 10 micromachines-14-01163-f010:**
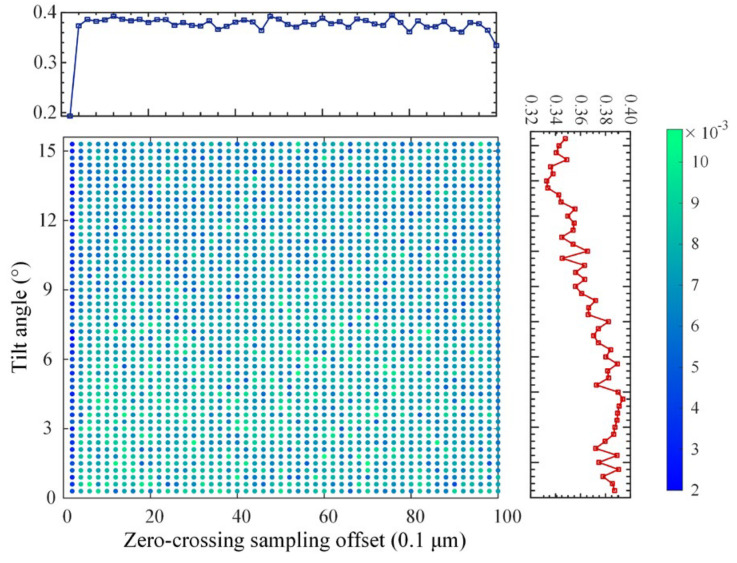
PCC error distribution.

**Figure 11 micromachines-14-01163-f011:**
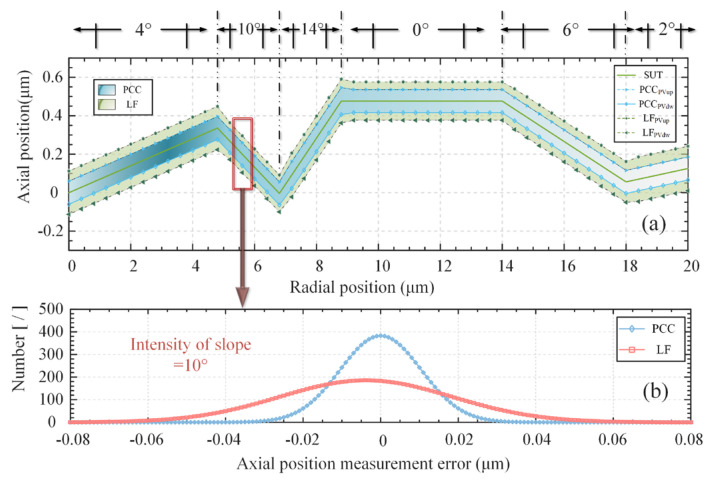
Comparison of algorithms in simulated experiments. (**a**) Overall measurement results. (**b**) Error distribution when the tilt angle is 10°.

**Figure 12 micromachines-14-01163-f012:**
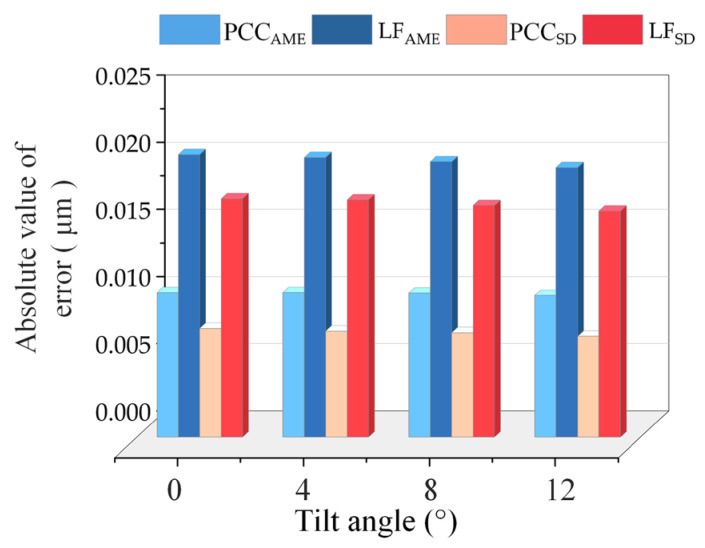
PCC compensation strategy and linear fitting (LF) error mean and standard deviation outcomes.

**Figure 13 micromachines-14-01163-f013:**
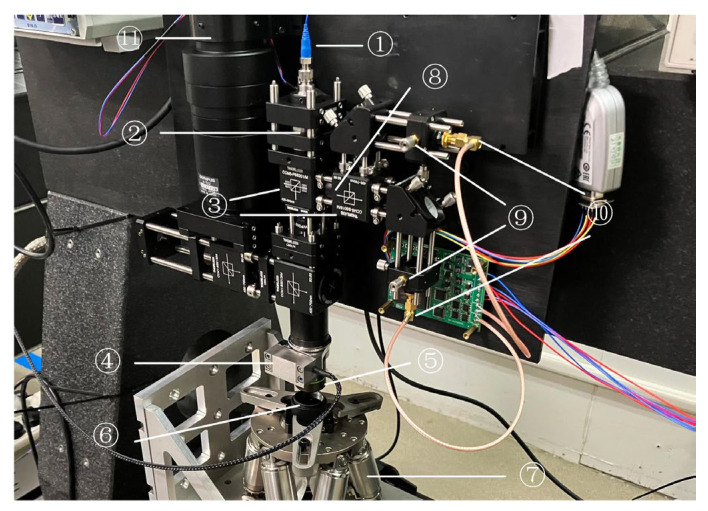
Physical structure of the DCM. (①: Laser; ②: Collimating lens; ③: Splitter; ④: PZT; ⑤: Objective; ⑥: Standard subject; ⑦: Six-axis displacement mechanism; ⑧: Focusing lens; ⑨: Pinhole; ⑩: Detector; ⑪: CCD).

**Figure 14 micromachines-14-01163-f014:**
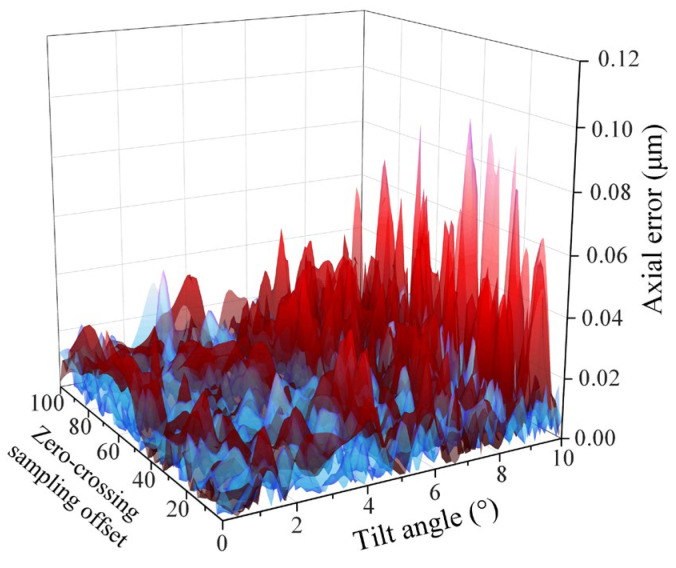
Comparison of measurement errors between PCC and LF.

**Figure 15 micromachines-14-01163-f015:**
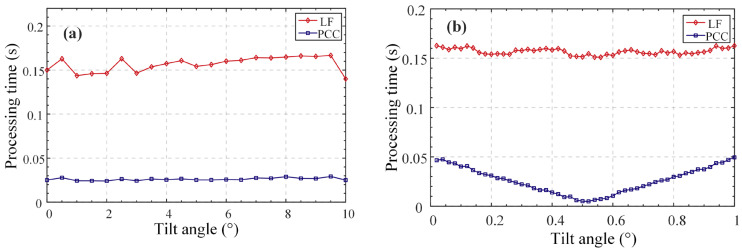
Comparison of computing efficiency of PCC and LF. (**a**). Comparison at different rotation depths. (**b**). Comparison of different sampling offsets.

**Figure 16 micromachines-14-01163-f016:**
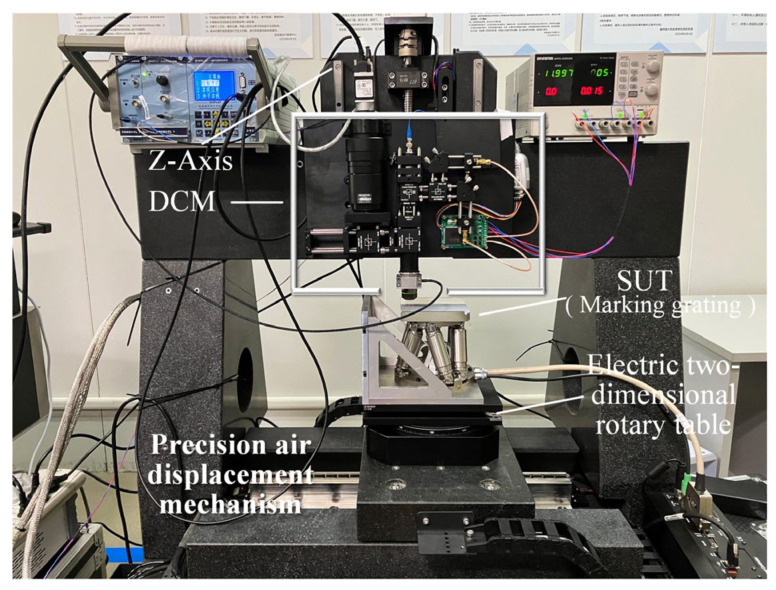
Three-dimensional measurement experimental system.

**Figure 17 micromachines-14-01163-f017:**
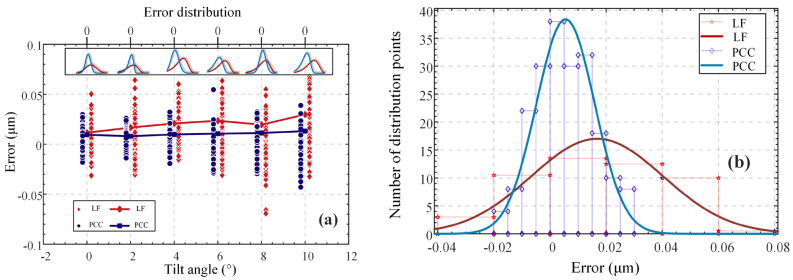
Error distributions of repeatable experiments. (**a**) Error distributions at different angles. (**b**) Error distribution of the system when the tilt angle is 0°.

**Figure 18 micromachines-14-01163-f018:**
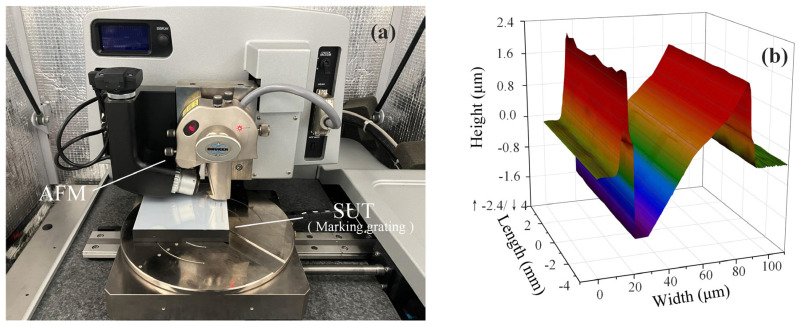
**Figure 18**. (**a**) Atomic force microscopy (AFM) was used to obtain true values. (**b**) The fitting results were obtained on the upper surface of the scratch grating with a nominal tilt value of 3°.

**Figure 19 micromachines-14-01163-f019:**
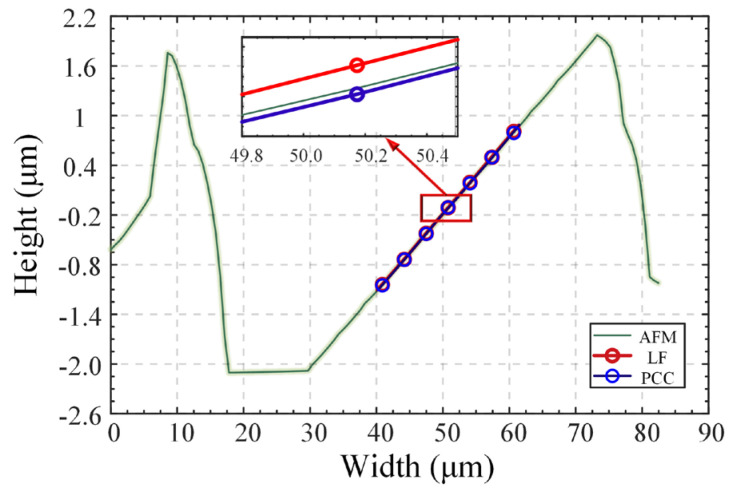
Random measurement results at the nominal tilt value of 6° on the upper surface of a scratched grating.

**Table 1 micromachines-14-01163-t001:** Variations of mean errors at different levels of noise.

Voltage	Mean Error of Pearson’s Correlation Coefficient	Mean Error of Linear Fitting
0 V	0.0104 μm	0.0155 μm
5 V	0.0103 μm	0.0155 μm
8 V	0.0109 μm	0.0167 μm
12 V	0.0126 μm	0.0244 μm

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
