# Peer review of "Differential Confocal Optical Probes with Optimized Detection Efficiency and Pearson Correlation Coefficient Strategy Based on the Peak-Clustering Algorithm"

_micromachines, 2023, doi:10.3390/mi14061163_

Round 1

Reviewer 1 Report

Quantifying free-form surfaces with balancing accuracy and efficiency is an important topic. The authors presented a detailed framework and validate the algorithms using simulations and experiments. The manuscript is well written with impressive graphs. There are a few improvements for the authors to consider.

(1) An experiment and simulation flowchart can be added at the end of the Introduction part. This will help readers to better grasp the main innovative technologies and ideas addressed in the manuscript.

(2) The manuscript focuses on the surface observation scanning techniques, but recent applications of advanced CT facilities in interior 3D measurement are also an vital aspect. Therefore, the authors need to comment on the 3D CT observation of the interiors of equipment or complex materials, which can refer to e.g. Int J Solids Struct, 2015, 67: 340-352, 10.1016/j.cemconcomp.2021.104347. This can serve to expand the scope of contemporary optical studies, industrial operations, and innovative materials.

(3) There are many equations involved, but please check carefully whether the subscripts are in italics or in regular font, for example, the h in Eq. (18).

(4) Figure 12, Page 14: The text descriptions in the figure are mixed with the equipment, making it inconvenient to read. The reviewer thus suggests the authors to replace them with numbered circles and provide explanations for these numbers outside the figure.

(5) Section 6, Page 19: Amendments are required to summarize the main conclusions in a point-by-point manner.

The manuscript is well written and need only minor revisions.

Reviewer 2 Report

The authors proposed an improved algorithm for optical probing. The data presented is solid and validated the authors claim. I have no further questions and recommend the manuscript to be published

Reviewer 3 Report

The paper is quite well-written and generally clear. It is somewhat difficult to follow the logic in the math part of the manuscript in Section 3, and I would suggest the authors revise it for better clarity and readability. However, it is optional. Still, the authors need to check parentheses in eqs. (12) and (13), and explain the meaning of P_h and P_v in Algorithm 1.   

In Fig. 2, the annotation for pinhole B is missing.  It would be also useful to indicate in the figure that the pinholes are before and after focus. 

The authors need to check English in the last paragraph of p. 7, starting with "For real-time acquisition... ". Probably, splitting the sentences into shorter sentences will help to improve readability. Now, the meaning of the paragraph is unclear.
